

# MicroRNAs tend to synergistically control expression of genes encoding extensively-expressed proteins in humans

Xue Chen, Wei Zhao, Ye Yuan, Yan Bai, Yong Sun, Wenliang Zhu and Zhimin Du

Department of Pharmacy, The Second Affiliated Hospital of Harbin Medical University (Institute of Clinical Pharmacy, The Heilongjiang Key Laboratory of Drug Research, Harbin Medical University), Harbin, China

## ABSTRACT

Considering complicated microRNA (miRNA) biogenesis and action mechanisms, it was thought so high energy-consuming for a cell to afford simultaneous over-expression of many miRNAs. Thus it prompts that an alternative miRNA regulation pattern on protein-encoding genes must exist, which has characteristics of energy-saving and precise protein output. In this study, expression tendency of proteins encoded by miRNAs' target genes was evaluated in human organ scale, followed by quantitative assessment of miRNA synergism. Expression tendency analysis suggests that universally expressed proteins (UEPs) tend to physically interact in clusters and participate in fundamental biological activities whereas disorderly expressed proteins (DEPs) are inclined to relatively independently execute organ-specific functions. Consistent with this, miRNAs that mainly target UEP-encoding mRNAs, such as miR-21, tend to collaboratively or even synergistically act with other miRNAs in fine-tuning protein output. Synergistic gene regulation may maximize miRNAs' efficiency with less dependence on miRNAs' abundance and overcome the deficiency that targeting plenty of genes by single miRNA makes miRNA-mediated regulation high-throughput but insufficient due to target gene dilution effect. Furthermore, our *in vitro* experiment verified that merely 25 nM transfection of miR-21 be sufficient to influence the overall state of various human cells. Thus miR-21 was identified as a hub in synergistic miRNA–miRNA interaction network. Our findings suggest that synergistic miRNA–miRNA interaction is an important endogenous miRNA regulation mode, which ensures adequate potency of miRNAs at low abundance, especially those implicated in fundamental biological regulation.

Corresponding author
Zhimin Du, dzm1956@126.com

## INTRODUCTION

MicroRNAs (miRNAs) belong to a super-family of ∼22 nucleotides single-stranded non-coding RNA molecules, which are extensively implicated in pathophysiological activities (*Bartel, 2009*; *Shukla, Singh & Barik, 2011*). After a long biogenesis process, functionalities of miRNAs rely on their interaction with argonaute proteins and co-assembling into the

RNA-induced silencing complex (RISC). Despite being energy-consuming, such a unique gene regulation pattern may be significant to human cells owing to their macro roles in biological regulation (*Adlakha & Seth, 2017*; *Park et al., 2016*; *Srinivasan & Das, 2015*).

One miRNA usually physically interacts with hundreds of target genes (*Zhou & Yang, 2012*), suggesting the advantage of high-throughput and integrative gene regulation (*Liu et al., 2016*). However, it was also proposed that an inherent deficiency should be not ignored regarding the operation of a 'one to many' system like this. Targeting hundreds of genes undoubtedly dissipates the efficacy of the miRNA-RISC machine due to the abundance dilution effect of target genes (*Arvey et al., 2010*). Undoubtedly, comprehensive and effective regulation covering all target genes needs high miRNA output with sufficient RISC machines in which mature miRNA is embedded. Such a strategy may be uneconomical for widespread adoption due to excessive occupancy of cellular energy and material. Actually, the completed human miRNA expression profiles have shown that only a very small portion of miRNAs can afford this way, such as miR-1 that is strongly expressed in heart and miR-122a that is highly expressed in liver (*Ritchie, Flamant & Rasko, 2010*). Just recently, it has been revealed that ∼10–15% of human miRNAs are tissue-specific, replying limited cell load capacity for miRNAs (*Ludwig et al., 2016*).

Indirect and non-physical interactions exist among miRNAs, which forms the basis of the overall effect of miRNA-mediated gene regulation. Several possible patterns of miRNA–miRNA interactions (MMIs) were uncovered by far. As one miRNA can physically interact with hundreds of target genes, this causes with a great probability that different miRNAs may competitively bind the 3′-untranslated region of the same gene. Competition for binding is the main and fundamental pattern of miRNA-miRNA interactions (MMIs) in the miRNA world (*Jens & Rajewsky, 2015*). Expect the competition pattern, different miRNAs may show cooperativity due to being co-regulated by the same transcription factor and targeting genes with functional interconnections (*Na & Kim, 2013*; *Shi et al., 2013*; *Guo et al., 2014*). Restricted MMI is another pattern that two miRNAs have completely or partially complementary structures and constitute an endogenous sense and antisense miRNA pair (*Guo et al., 2014*). Great expression difference of the two miRNAs in a pair was a major feature of this pattern.

Based on our previous knowledge of miRNA regulation (*Yuan et al., 2015*; *Zhu et al., 2011*; *Zhu et al., 2013*), we proposed that miRNAs might adopt an alternative pattern of MMI that promotes a manner of more economical and efficient gene regulation in response to real-time adjustment of cellular signals. It less depends on miRNA abundance and more rely on synergistic miRNA-miRNA collaboration for fine-tuning protein output (*Skommer et al., 2014*). On the whole, synergistic gene regulation may optimize the regulatory efficacy of miRNAs (*Xu et al., 2011*). Importantly, it is less expensive in cellular energy consumption compared to extraordinarily high expression of miRNAs. In order to validate this hypothesis, we performed a quantitative assessment of miRNA synergism by calculating miRNA synergy score (*Zhu et al., 2013*). Benefited from the increasingly clear human protein expression atlas (*Uhlén et al., 2015*), this method enables us to explore potential principles in miRNA regulation at the dimension of human organs. Additionally, *in vitro* experiment was performed to show the potential benefit of miRNA synergy. Such efforts
are intended to shed novel insights into the biological significance of miRNA-miRNA collaboration and provide implications for better understanding the existence of miRNAs in humans.

## MATERIALS AND METHODS

### MiRNA-target interactions (MTIs) and protein expression

Three datasets (miRecords (*Xiao et al., 2009*), miRSel (*Naeem et al., 2010*) and ExprTargetDB (*Gamazon et al., 2010*)) were used to obtain reliable MTIs in humans. MiRSel is a collection of literature evidence of MTIs. MiRecords and ExprTargetDB belong to secondary tools for MTI prediction, in which integration of different algorithms led to reliable MTI identification. Finally, all MTI data were merged for further analysis. Organ protein expression data were obtained from the human protein atlas (HPA) database (*Uhlén et al., 2015*). Twelve human organs were included: breast and female reproductive system (BFS), blood and immune system (BIS), central nervous system (CNS), cardiovascular system (CVS), digestive tract (DT), endocrine glands (EG), liver and pancreas (LP), male reproductive system (MS), placenta (P), respiratory system (RS), skin and soft tissues (SS) and urinary tract (UT). A resilient fraction threshold was adopted for expression verification. Specifically, the fraction threshold was set at 70% for BFS, BIS, CNS, LP and SS; 90% for DT; 100% for CVS, EG, MS, P, RS and UT. The information of the level of annotated protein expression was downloaded and imported into Cytoscape v2.8.3 (*Smoot et al., 2011*). Notably, validated marks 'none', 'low', 'medium' and 'high' representing proteins expression levels were in advance converted into the digitals 0, 1, 2 and 3, respectively. Afterwards, coefficient of variation (CV) was used to assess the cellular expression dispersion degree of each miRNA target gene-encoded protein.

### Synergy score and skewness

Potential degree of synergistic collaboration between miRNAs was quantitatively assessed by calculating miRNA synergy score as described before (*Zhu et al., 2013*). For each miRNA, the statistics parameter skewness was used to assess the distribution inclination of the $\log_2 CV$ values of protein expression and determine the regulatory tendency of the miRNA (*Mardia, 1970*). We used $\log_2 CV$ instead of CV for skewness calculation, as the distribution of CVs did not pass D'Agostino & Pearson omnibus normality test (*D'Agostino & Pearson, 1973*). A smaller skewness value means a tendency of regulating mRNAs encoding uniformly expressed proteins (UEPs); however a higher skewness value suggests that a miRNA tends to fine-tune disorderly expressed proteins (DEPs). File S1 was a four-step protocol to reveal miRNAs that tend to regulate UEP-encoding mRNAs. In this study, UEPs and DEPs refer to proteins with CVs of less than 40% and CVs of more than 120%, respectively.

### Gene ontology (GO) and network topology

A comparison between UEPs and DEPs was investigated from the GO aspect by applying the online DAVID functional annotation tool (*Huang da, Sherman & Lempicki, 2009*). Briefly, gene official symbols were submitted as gene list and 'Homo Sapiens' was selected

as the background. Over-represented GO biological processes, cellular components and molecular functions were considered to be significant only if false discovery rate (FDR) was less than 0.05. We also investigated network topology characteristics of UEPs and DEPs, respectively. The Cytoscape plugins BisoGenet (*Martin et al., 2010*) was used to retrieve, trim, and analyze experimentally validated protein-protein interactions (PPIs).

## Cell culture and miRNA transfection

Human umbilical vein endothelial cell line (HUVEC), breast cancer cell line (MCF7) and hepatic carcinoma cell line (HepG2) were all obtained from the Institute of Biochemistry and Cell Biology, Chinese Academy of Science (Shanghai, China). Cells were maintained in Dulbecco's modified Eagle's medium (DMEM) supplemented with 10% fetal bovine serum, 100 U/mL penicillin and 100 μg/mL streptomycin. After that, cells were incubated at 37 °C with 5% $CO_2$ and 95% air. 24 h before transfection, cells were transferred to 96-well plates and cultured in fresh medium without antibiotics, According to the manufacturer's instructions, X-treme GENE siRNA transfection reagent (Roche, Basel, Switzerland) was used for mono-transfection of negative control or miRNA mimic at 25 nM. Table S1 listed the sequences of the negative control and miRNA mimics used in this study.

## Cell viability assay

Cell viability was assessed by measuring mitochondrial dehydrogenase activity, using the colorimetric MTT assay, based on the fact that viable cells (but not dead cells) can reduce 3-(4,5-dimethylthiazol-2-yl)-2,5-diphenyl tetrazolium bromide (MTT). After miRNA transfection 24 h, with or without hypoxia treatment, cells were incubated with MTT of 5 mg/ml at 37 °C for four hours. Hypoxia was induced by exposing cells to hypoxia condition (1% $O_2$, 94% $N_2$, 5% $CO_2$) for 24 h using a modular incubator. The purple formazan crystal was dissolved with 150 μL of dimethyl sulfoxide (DMSO) and added to the cells. The absorbance was measured at 490 nm.

## Data statistics

All data are expressed as mean ± SEM (Standard Error of Mean). Statistical analysis was performed with Mann–Whitney *U* test or one-way ANOVA (analysis of variance) followed by Tukey's test for multiple comparisons. Differences were only considered to be significant at $p < 0.05$.

# RESULTS

## Data collecting and screening

Data integration identified a total of 11,162 target genes and 69,618 MTIs for 472 miRNAs from the three databases including miRecords, miRSel, and ExprTargetDB. Only 166 of were retained for further analysis, each of which targeted at least 50 mRNA genes in each of the 12 human organs (Table S2). Totally, they were functionally associated with more than 6,328 proteins via 36,211 MTIs.
## Comparison of UEPs and DEPs in organ expression, network topology and GO terms

After retrieving the protein expression data from the HPA database, we calculated the CV value for each of the 6,328 proteins regarding its HPA-defined abundance ranks in human cells. The majority of the proteins were located with the CV range from 40% to 120% (Fig. 1A). This distribution characteristic was more clearly observed after log2 transformation of CVs. In total, we identified 1,340 UEPs and 1,115 DEPs. Unlike DEPs, UEPs tend to be more uniformly expressed in human organs, have more interacting neighbors, and constitute closer associations with each other in clusters (Figs. 1B–1D). Comparably, DEPs are not ubiquitously expressed, and more marginalized in the human PPI network and execute functions in isolation. More than these, GO analysis disclosed that UEPs tend to be intensively involved in gene transcription and related biological processes, be located in the nucleus or nearby, and play roles in the housekeeping functions such as being transcription regulators (Table S3). This result indicates the crucial role of UEPs in the maintenance of basic cell activities. In contrast, DEPs are more implicated in organ-specific biological processes and perform molecular functions at the plasma membrane and cytoplasmic organelles.

## MiRNAs' regulation tendency on mRNAs encoding UEPs or DEPs is independent of miRNA expression

We investigated the difference of UEPs and DEPs from the miRNA regulation aspect. UEP-encoding mRNAs instead of those encoding DEPs are averagely under more dense control of miRNAs in the whole genome context (Fig. 2A). Even when MTIs are restricted into one organ such as CVS, preference of miRNAs for UEP-encoding mRNAs is also obvious to observe (Fig. 2B). The highest skewness value highlighted miR-133a of its definite trend in regulating DEP-encoding mRNAs in CVS (Fig. 2C). Both miR-1 and 133a are cardiac-specific miRNAs. However, we found that they showed different tendencies on gene regulation (Fig. 2D). This result suggests that regulating organ-specific genes may be not just a necessary condition for the organ-specific functions of miRNAs, such as miR-1 (Yang et al., 2007).

## MiRNAs regulate UEP-encoding mRNAs in a synergistic pattern

MiRNAs may act synergistically with each other (Xu et al., 2011). With the evolution of the complexity of a biological system, synergism is undoubtedly more advantageous than isolated regulation in terms of management strategy (Corning, 1995; Stelling et al., 2004). Altered expression of only a few numbers of miRNAs caused systemic changes via disrupting synergistic associations between miRNAs (Figs. 3A–3C). If all conceivable miRNA pairs are supposed to be synergistic 75.5% of synergistic miRNA interactions will be affected only when half of all the miRNAs undergo altered expression (Fig. 3D). In particular, to those the hubs in the miRNA–miRNA synergistic network, adaptation of this management strategy may be more energy-saving due to plenty of synergistic partners (Fig. 3E). Further result indicates that miRNAs with small skewness values comprise most of the synergistic miRNA–miRNA associations in human organs (Fig. 4). This finding implies that the molecular behaviors of UEPs, rather than those of DEPs, are under the surveillance of a

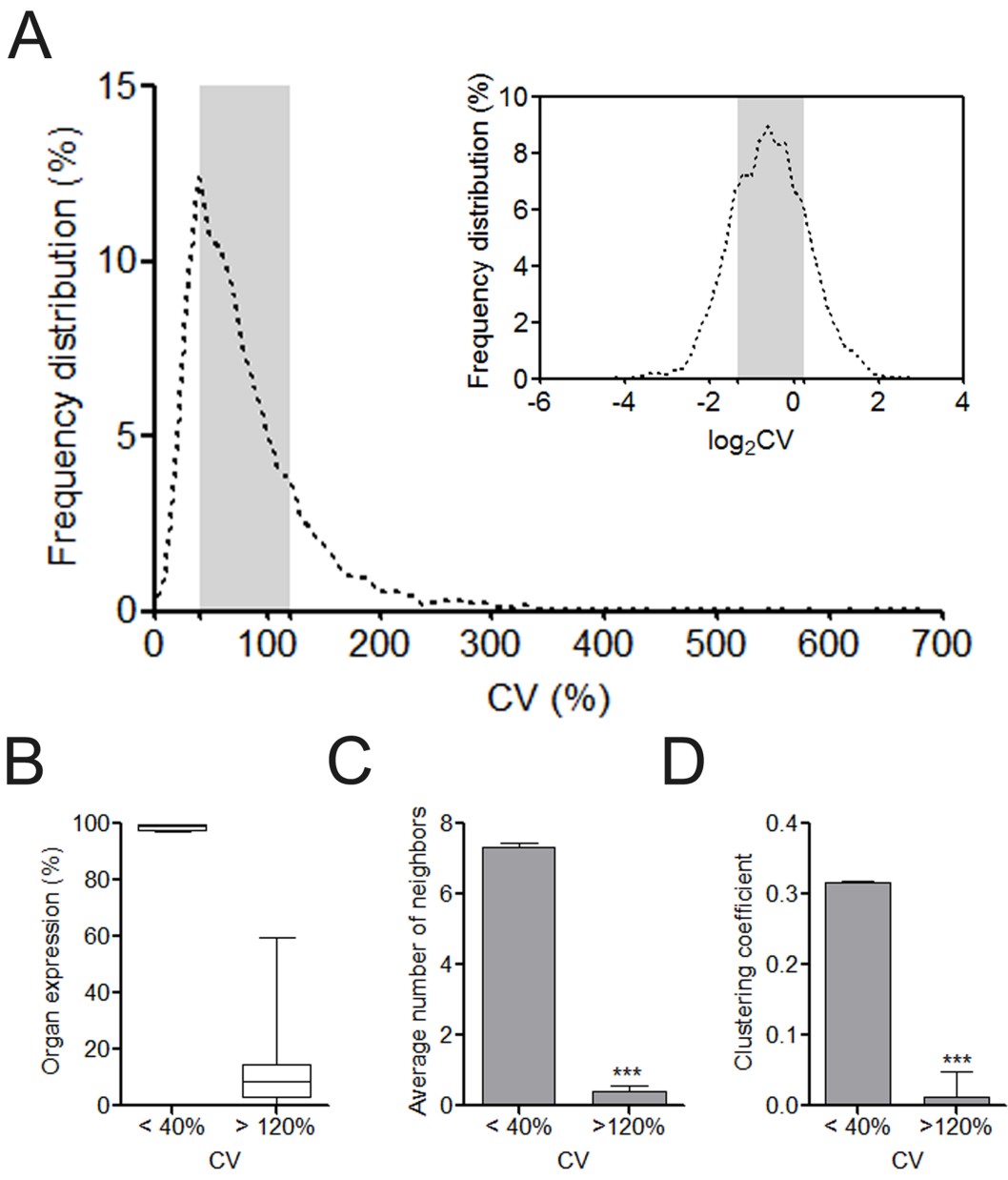

**Figure 1** **Overview of organ expression of proteins encoded by miRNA target.** (A) Frequency distribution of coefficient of variation (CV) of protein expression abundance in 12 human organs. The right upper showed the frequency distribution of $\log_2 CV$. The CV interval from 40% to 120% was highlighted with gray box. (B) The box and whiskers plot of organ expression percentage of UEPs and DEPs. (C) Comparison of average number of neighbors of UEPs and DEPs. ***$p < 0.001$, UEPs versus DEPs; (D) Comparison of cluster coefficient of UEPs and DEPs. ***$p < 0.001$, UEPs versus DEPs. UEPs: uniformly expressed proteins; DEPs: disorderly expressed proteins.

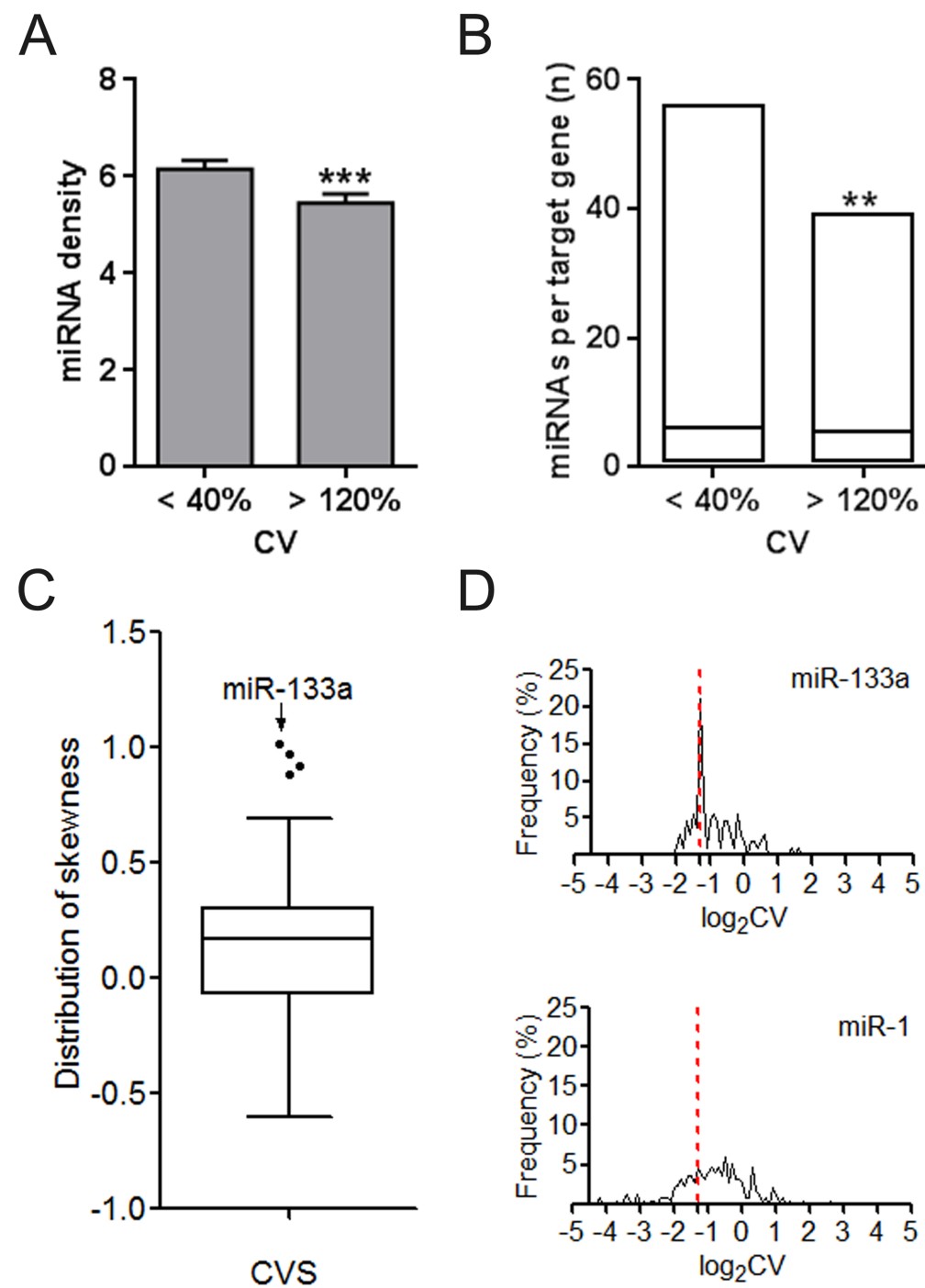

**Figure 2 Regulation density and tendency of miRNAs on UEP-encoding genes and DEP-encoding genes.** (A) Comparison of miRNA density on UEP-encoding genes and DEP-encoding genes. The calculated miRNA density refers to miRNA target sites within only 3′ untranslated regions of genes. (B) Distribution of the number of miRNAs per target genes that encode proteins expressed in cardiovascular system. **$p < 0.01$, ***$p < 0.001$, UEP-encoding genes versus DEP-encoding genes. (C) Skewness assessment of miRNAs in cardiovascular system (CVS). (D) Distribution of expression $\log_2$CVs of proteins encoded by miR-133a and miR-1 target genes. The red dotted line at $\log_2$CV of $-1.32$ corresponds to the CV value of 40%. CV: coefficient of variation. UEPs: uniformly expressed proteins; DEPs: disorderly expressed proteins.

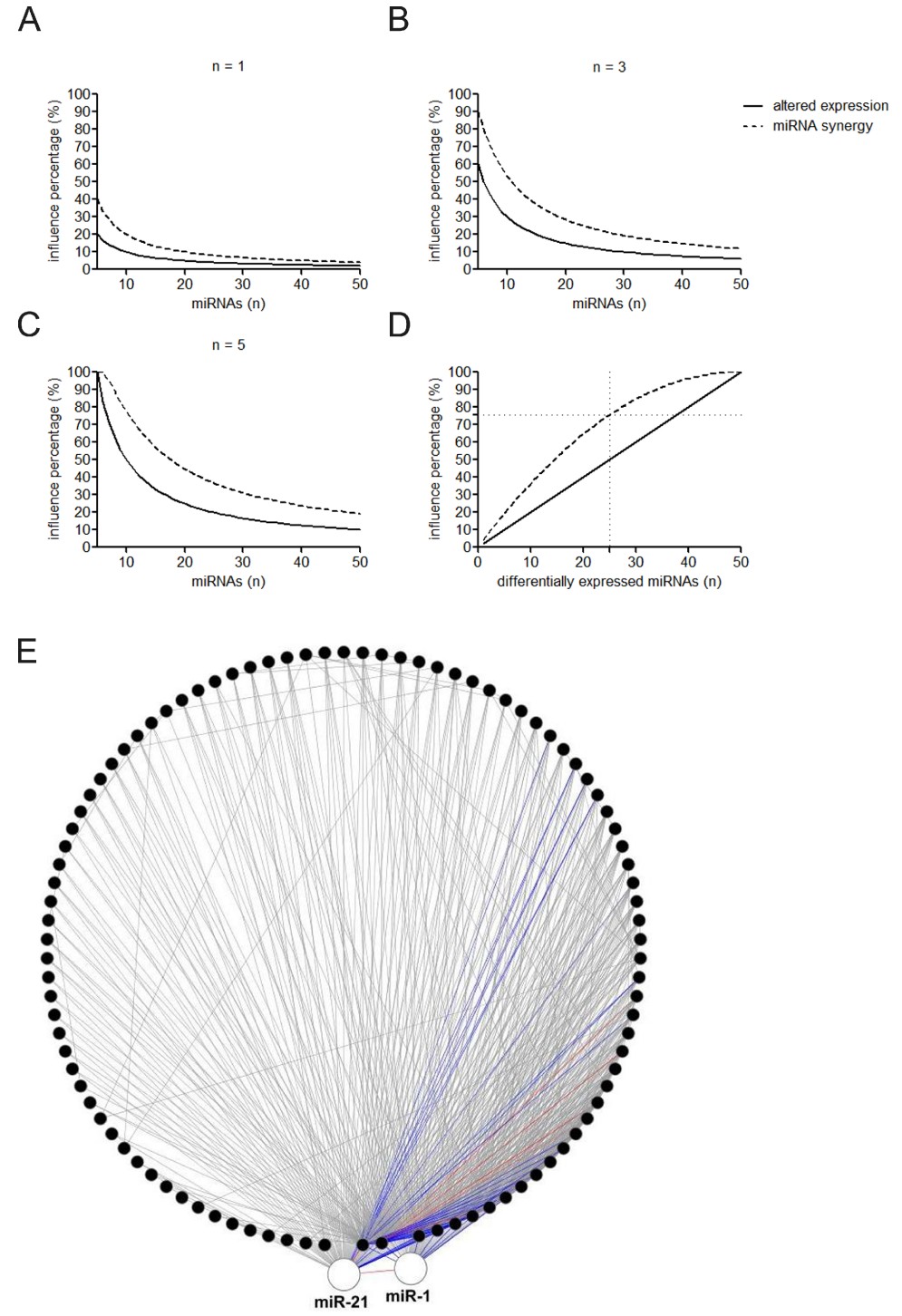

**Figure 3  Influence of altered expression and synergy on the overall miRNA regulation.** (A–C) Percentage of the affected miRNAs and synergistic miRNA interactions at different number of differently expressed miRNAs. $n = 1$, 3, or 5 means that one, three, or five miRNAs are simultaneously dysregulated. (D) Percentage of the affected miRNAs and synergistic miRNA interactions by altered miRNA expression. (E) Synergistic miRNA-miRNA interactions in cardiovascular system. The red, blue and gray lines represent interactions with synergy scores of >2.0, >1.5 and >1.0, respectively. Two hubs in the synergistic miRNA interaction network miR-1 and miR-21 were highlighted as big nodes.

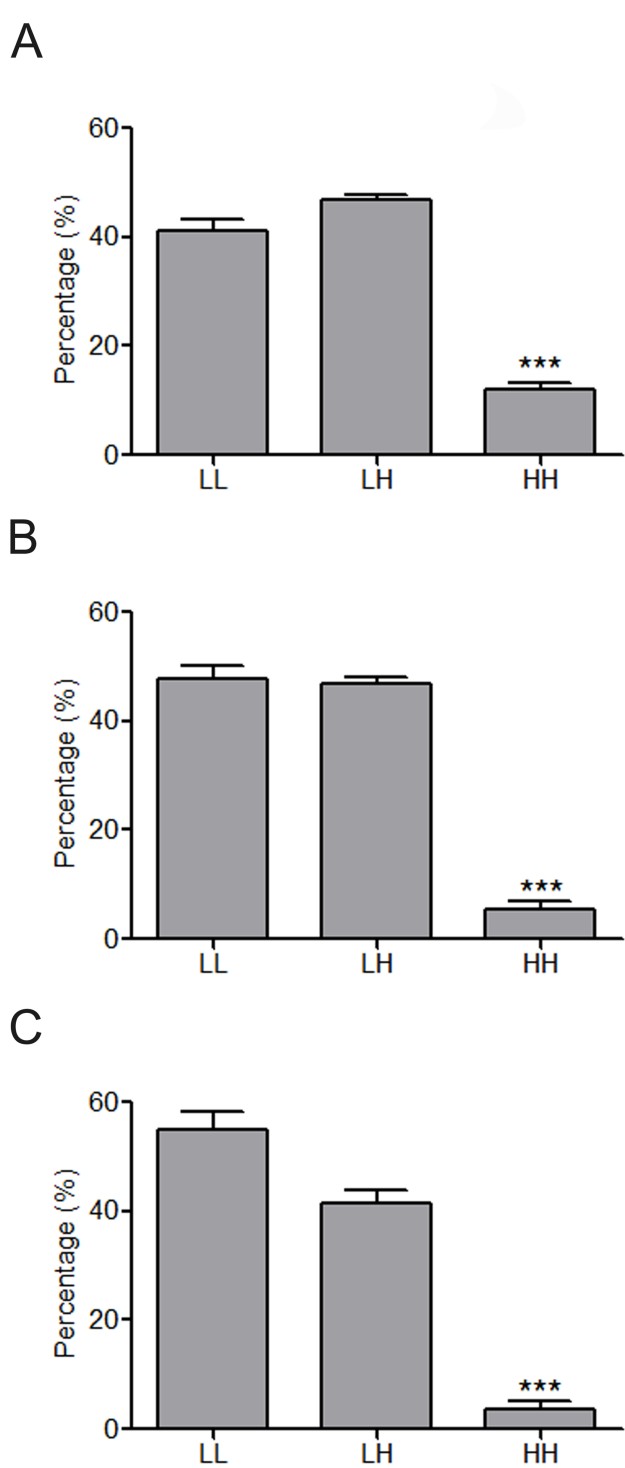

**Figure 4** **Constitution share of low and high skewness miRNAs for top 500 (A), top 100 (B) and top 50 (C) synergistic miRNA interactions in 12 human organs.** \*\*\*$p < 0.001$, HH versus LL; LL, LH and HH represent synergies between low skewness miRNAs, between low skewness miRNAs and high skewness miRNAs, and between high skewness miRNAs, respectively.

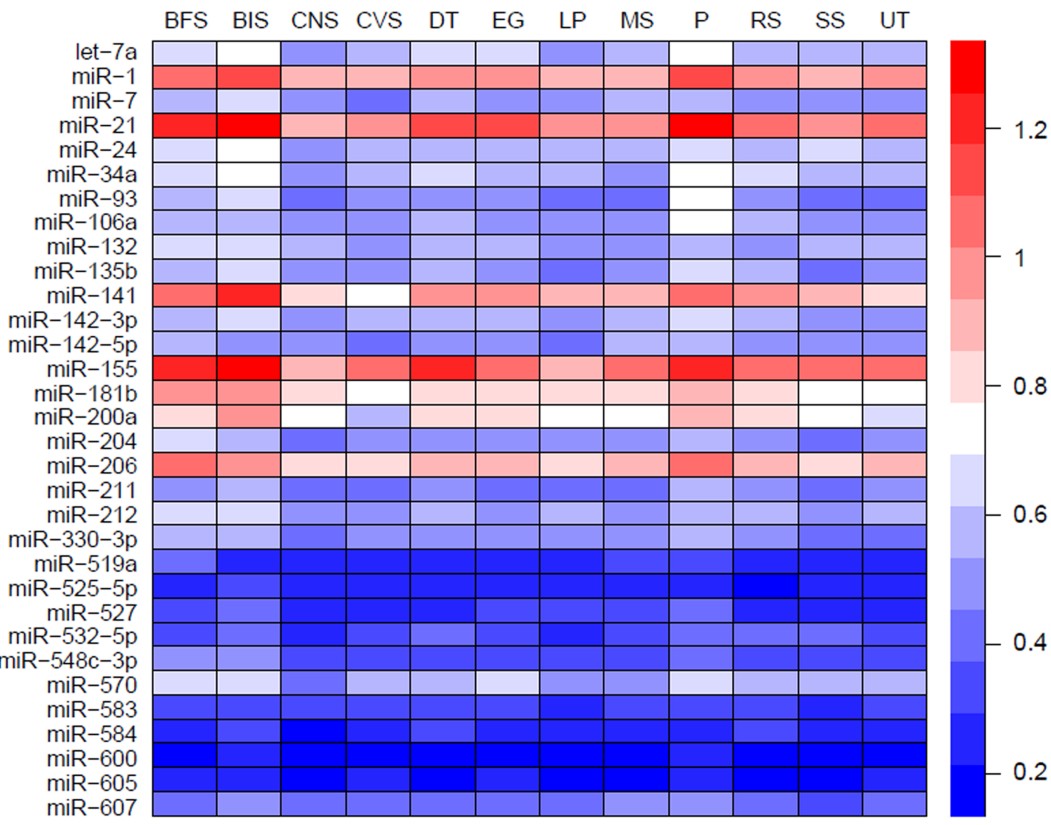

**Figure 5  Heatmap of mean synergy scores of miRNAs with skewness values of <0.3 in 12 human organs.** BFS, breast and female reproductive system (female tissue); BIS, blood and immune system (hematopoietic); CNS, central nervous system (brain); CVS, cardiovascular system (heart and blood vessels); DT, digestive tract (GI-tract); EG, endocrine glands; LP, liver and pancreas; MS: male reproductive system (male tissues); P, placenta; RS, respiratory system (lung); SS, skin and soft tissues; UT, urinary tract (kidney and bladder).

dense synergistic miRNA regulation network. In this invisible network, several miRNAs should be paid more attention owing to their more powerful and extensive collaboration with other miRNAs (Fig. 5). They are miR-1, miR-21, miR-141, miR-155, miR-181b, miR-200a, and miR-206. Compared with other miRNAs, miR-21 had a significant impact on the whole cell state of the three cell lines at a mono-transfection concentration of 25 nM (Fig. 6). MiR-133a tends to regulate DEP-encoding mRNAs implying its relatively isolated action, our *in vitro* experiment verified its failing to influence the whole cell state of the three cell lines.

## DISCUSSION

The new discovery of miRNAs and other kinds of non-coding RNAs greatly consolidates the dominant position of genes in regulating cellular activities (*Cech & Steitz, 2014*). Genes are not only carriers and messengers of genetic information, but also direct supervisors on whether such information is accurately translated into phenotypes. Despite many efforts it is not still clear about the regulation layer that is constituted by thousands of miRNAs by

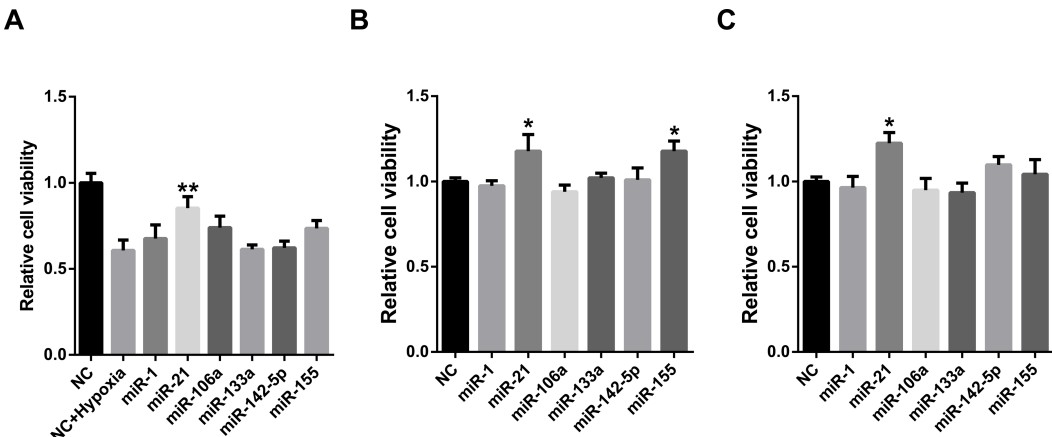

**Figure 6** Cell viability assay of miRNA transfection on cell lines of HUVEC (A), MCF7 (B), and HepG2 (C). $**p < 0.01$, versus NC + Hypoxia (HUVEC); $*p < 0.05$, versus NC (MCF7 and HepG2); NC, negative control; $n = 6$.

now (*Boettger & Braun, 2012*; *Stroynowska-Czerwinska, Fiszer & Krzyzosiak, 2014*). In this study, system-level insight was attempted to shed into the overall effect of miRNAs from the perspective of miRNA–miRNA collaboration.

It was confirmed that abundant expression of target genes produces the abundance dilution effect for miRNA-mediated regulation (*Arvey et al., 2010*). Denzler and his colleagues *(2014)* further proved that the overall abundance of target genes is generally higher than that of the corresponding miRNA. This causes insensitive influence of competitive endogenous RNA (ceRNA)-mediated derepression on miRNA activity (*Denzler et al., 2016*). The above findings imply that miRNAs may do not perform functions in isolation and collaboration of different miRNAs is a means of effective regulation of gene expression.

Functional studies have confirmed that miRNAs might act synergistically, strongly suggesting collaborative gene regulation (*Huang et al., 2016*; *Raut et al., 2016*; *Xue et al., 2016*). The above evidence is properly consistent with the conclusion inferred by the systems biology analyses performed before (*Xu et al., 2011*; *Zhu et al., 2013*). Extensive miRNA–miRNA synergism may exist in the miRNA layer. Precisely because of this, slight adjustment of miRNAs' abundance can cause an apparent variation in the state of cells. Synergistic miRNAs action allows cells to adjust their live-or-death status more flexibly and rigorously to adapt to internal and external alternation. Due to acting synergistically with other miRNAs, low concentration transfection of miR-21 showed an obvious impact on the viability of all of the studied cell lines. Our result is in agreement with the established knowledge that miR-21 is a well-known miRNA that targets tumor suppressor genes and was shown experimentally in many studies to control apoptosis (*Buscaglia & Li, 2011*). This finding also implies that altered miRNA expression may not only independently affect the translation of its own target gene products but also produces a radiative impact on other miRNAs that have synergistic associations with it. Put briefly, miRNA synergy amplifies the effects of miRNA expression alteration. As cell viability assay is only one way of measuring effects of miRNAs on the cell state and it cannot be excluded that the other

examined miRNAs with high potential to regulate UEP-encoding mRNAs can affect the cell state in another way. However, further studies are definitely needed to investigate the miRNAs through other experimental methods.

Functional execution of miRNAs requires high cellular energy consumption as a prerequisite (*Shukla, Singh & Barik, 2011*). Besides miRNA regulatory crosstalk (*Jens & Rajewsky, 2015*), miRNA synergy may represent an alternative pattern of energetically optimal miRNA-mediated regulation of post-transcriptional gene expression. However, we supposed that this obvious advantage brought by miRNA synergy might rely heavily on the complexity of the biological system. As miRNA synergy score calculation disclosed, only dense PPIs between target gene products could lead to meaningful miRNA synergy (*Zhu et al., 2013*). This explains well that one can expect strong synergistic potential of the cardic-specific miRNA miR-1 instead of miR-133a. Our findings indicate that miR-1 has a tendency to regulate UEP-encoding mRNAs whereas miR-133a tends to regulate DEP-encoding mRNAs. A deeper insight is that only complex life forms such as human can afford the coexistence of hundreds of miRNAs in a cell owing to the intricate PPI network that is an essential condition for miRNA synergy to occur.

Besides, our results further point out that miRNA synergism may be associated with the organ expression of target gene products. UEPs that have broad and balanced expression in human organs are subject to over-supervision by synergistic miRNA regulation. Broad and balanced expression of UEPs is a reasonable reflection of their important biological roles in cellular activities such as significant participation in gene transcription. Uniformed expression of large number of proteins throughout human organs implies that different cells share a similar basic state, which ensures that the communication between cells is relatively equal in the physiological environment. Strengthened synergistic miRNA regulation on UEP-encoding mRNAs suggests that miRNA synergism should contribute to the maintaining of their uniformed expression throughout human organs. Due to allowing cells more finely tune the protein abundance of UEPs, miRNA synergism leads to more economical adaptation of cells to the intracellular and extracellular environments.

The complexity of the biological system is described by the following characteristics such as robustness, redundancy, and crosstalk (*Jia et al., 2009*). Biological robustness lies in that altered expression of a single gene may be insufficient to effectively affect the overall system (*Kitano, 2004*). Although this guarantees the relative stability of a biological system, it makes cells difficult to adapt to the external environment changes in a timely manner. Due to the existence of functional linkages between system components, cells cannot merely change one component but keep others unchanged when faced with altered conditions. For instance, as cells encounter excessive oxidative stress, adjustment of the overall cellular signals could be observed (*Chandra, Samali & Orrenius, 2000*; *Kiffin, Bandyopadhyay & Cuervo, 2006*; *Reuter et al., 2010*). Robustness and adaptability are mutually contradictory. Biological system requires an integrated gene/protein management strategy for coordinating diverse biological signals. By targeting hundreds of mRNA messengers, single miRNA is in a position to accomplish integrated gene regulation (*Backes et al., 2017*). Compared to mRNA-mediated protein expression control, indirect regulation by miRNAs is high-throughput (*Baek et al., 2008*). This may be the biological significance

of miRNAs' presence in human biology. More importantly, the unique synergistic action mechanism of miRNAs provides richer control skills for cells and better overcomes the huge energy consumption in biosynthesis of miRNAs and assembly of RISC protein complex.

Undoubtedly, more in-depth exploration of constituent factors for effective miRNA target site (*Agarwal et al., 2015*) would greatly promote our understanding of synergistic interactions between miRNAs. Differential 3′-untranslated region isoforms may lead to inconsistent target gene profiles of a miRNA in different types of cells (*Nam et al., 2014*). Combined with the fact that ∼10–15% of human miRNAs are tissue-specific (*Ludwig et al., 2016*), this finding implies that synergistic collaboration between miRNAs may be cell-type-specific or tissue-specific. However, more experimental researches are definitely to explore this.

In conclusion, no longer depending on self-expression lets miRNA synergism maximize the effectiveness of fine-tuning protein output. For complex biological systems, miRNA synergism seems to be a very energy-economical solution by which cells can better deal with the contradictory relationship between system robustness and system adaptability. Our findings provide a new understanding of the biological significance of miRNAs in the organ scale.

### Funding
This work was supported by the Major Program of the National Natural Science Foundation of China (81230081). The funders had no role in study design, data collection and analysis, decision to publish, or preparation of the manuscript.

### Grant Disclosures
The following grant information was disclosed by the authors:
Major Program of the National Natural Science Foundation of China: 81230081.

### Competing Interests
The authors declare there are no competing interests.

### Author Contributions
- Xue Chen and Ye Yuan reviewed drafts of the paper.
- Wei Zhao and Yan Bai performed the experiments.
- Yong Sun analyzed the data.
- Wenliang Zhu conceived and designed the experiments, analyzed the data, contributed reagents/materials/analysis tools, wrote the paper, prepared figures and/or tables.
- Zhimin Du contributed reagents/materials/analysis tools, reviewed drafts of the paper.

### Data Availability
The raw data has been supplied as a Supplementary File.

## Supplemental Information

Supplemental information for this article can be found online at http://dx.doi.org/10.7717/peerj.3682#supplemental-information.

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
