# Peer review of "MicroRNAs tend to synergistically control expression of genes encoding extensively-expressed proteins in humans"

_PeerJ, doi:10.7717/peerj.3682_

## Round 0.1 · original submission · Major Revisions

· Academic Editor

Major Revisions

Your manuscript has been carefully revised by four external reviewers. We really appreciate the attention to such an important and exciting topic. However, major revisions are requested to solve a number of methodological concerns, as well as to properly cite relevant missing references in the introduction, and address the criticisms related to the experimental part. Also, please pay attention in your revision to the English form and have a professional native English writer to help you out with the proof-reading. We would be glad to consider a revised version of the manuscript where all the points raised by the reviewers will be addressed.

·

Basic reporting

No comment

Experimental design

No comment

Validity of the findings

No comment

Comments for the author

The main idea of this article is a hypothesis on alternative miRNA
regulation with energy-saving and precise protein output. Using both
bioinformatics and experimental approaches, the authors suggest that
the synergistic miRNA-miRNA interaction is an important mechanism for
the regulation of protein levels. I believe that this article will shed
light on our understanding of a differential regulation of tissue-specific
genes. Such investigations provide an important contribution to
comprehension of mechanisms of pathological processes.

Reviewer 2 ·

Basic reporting

no comment

Experimental design

no comment

Validity of the findings

no comment

Comments for the author

Chen et al. reported that expression skewness of proteins encoded by miRNAs' target genes was evaluated at human organ scale, followed by quantitative assessment of miRNA synergism. This study may be interesting in the small RNA study, especially in miRNA-miRNA interaction. However, there are several concerns:
1. Generally, the phenomenon of miRNA-miRNA interaction is more widespread in the miRNA gene family and cluster, because these homologous miRNAs or clustered miRNAs are prone to detect close functional and evolutionary relationships. This study rarely mentioned these. Whether other dependent miRNAs can synergistically control target mRNAs, and the phenomenon is quite common in the small RNA world?
2. Indeed, miRNAs have been identified as regulatory molecules to target mRNA, and then the relevant protein level is inhibited. That is to say, miRNAs directly regulate target mRNAs, and then indirectly influence the final protein levels. Then, the title and some descriptions in this manuscript may be not appropriate.
3. For many researchers, the design is novel, and therefore it is quite essential to provide a flowchart for this study.
4. Except for synergistically regulation patterns, another miRNA-miRNA interaction should be not ignored: restricted interaction and competed interaction. I strongly suggested the authors should discuss it.
5. The relevant references have not properly cited. Many relevant references are not cited, and the introduction section is not enough to give a clear summarize and research aim.
6. Moreover, the English language is not of sufficient quality for publication. Please seek the help of an English language native speaker.

Reviewer 3 ·

Basic reporting

Overall, the manuscript is not well written concerning English language. It should be checked by native speaker.
Some sentences are not clear (for example, first sentence of the abstract). The term used: “expression skewness” is not common and could be misleading or not clear.

More papers should be cited which refer to miRNA mechanisms of action, as well as postulated synergy of miRNAs and repertoire of their targets (from Bartel group). Also in the Discussion section the authors should refer to ceRNA hypothesis.

More detailed description of the results should be included in the Results section, and NOT in the Figure Legends. This especially refers to Figure 3 and Figure 6.

Experimental design

If the authors want to support their conclusions with experimental data, more examples of miRNAs should be included. The presented data are somehow inconsistent:
- Why miR-21 mimic causes upregulation of cell viability in one case (Figure 6B)?
- Are control cells (Ctrl) treated with control oligonucleotide or non-transfected?

Validity of the findings

There are obvious misinterpretations, for example: in the legend of Figure 2B it is stated: “No significance was found between UEPs and DEPs”, whereas in the Results section these data are interpreted as: “the preference of miRNAs for UEPs is also obvious to observe”.

The authors could refer to their conclusions from their previous studies, for instatce: “MiRNAs that regulate more commonly expressed proteins also affect expression of more tissue-specific proteins”. Is it somehow inconsistent with presented data and conclusions in this manuscript?

The authors should refer to similar studies: “Understanding cooperativity of microRNAs via microRNA association networks” BMC Genomics 2013, “Walking the interactome to identify human miRNA-disease associations through the functional link between miRNA targets and disease genes” BMC Systems Biology 2013

Comments for the author

The thesis of this study should be more justified, as well as interpretation of the data more clear.

Minor comments:
- Typos: Lane 93, “Sampiens”, lane 224 – something is wrong with this sentence.
- Lane 169: “Genes (e.g. DNA, mRNAs) (…)” must be changed, for instance for: Genes and their expression products.
- Lane 171/172: Rewrite sentence: “Little is known (...)” because lots of studies were performed to reveal “regulation layer” of miRNAs. It is better to say “Despite lot of efforts still it is not clear…”. And include more general paper concerning activity of miRNAs, for instance: Stroynowska-Czerwinska et al. Cell Mol Life Sci. 2014

Reviewer 4 ·

Basic reporting

• The authors extensively talk in the introduction about miRNA synergistic interactions from the system biology perspective, which is done very clearly. However, I feel that the authors should mention miRNA competition between target sites, as an important part of miRNA regulatory crosstalk (reviewed in Jens and Rajewsky Nat Rev Genet 2015) and consider it in their discussion about energetically optimal miRNA-mediated regulation of gene expression.
• The figures are of good quality, meaningful and easy to understand. Minor adjustments needed:
o Figure 2A: the authors should specify whether the calculated miRNA density refers to experimentally validated/predicted miRNA target sites within only 3’ UTRs of genes or also 5’ UTRs and protein-coding regions.
o Figure 3A-C: the meaning of ‘n=1’, ‘n=3’, ‘n=5’ should be explained in the legend.
• Lines 46-47: a recent study suggests that around 10-15% of human miRNAs are tissue-specific (Ludwig et al. Nucleic Acid Res 2016). These findings should be included.
• Line 79: ‘each protein that were identified of being encoded by miRNA target genes’ – did the authors mean ‘regulated’?
• Line 133: ‘UEPs are not…’ – I think the authors meant ‘DEPs are not…’
• Line 138: the authors should avoid using ‘distal end of the nucleus’. My suggestion: ‘plasma membrane and cytoplasmic organelles’.
• Line 169: ‘Genes (e.g. DNA, mRNAs)’ should be rephrased.
• The text is well written and mostly clear. The English language is correct with minor mistakes, e.g. line 35 (‘may be of’), line 43 (‘covering on’), line 135 (‘in the nucleus’), line 176 (‘these evidence’), line 196 (‘such as human’).

Experimental design

• The authors have a proven track of publications focused on broadening our understanding of miRNA synergy and its effect on regulating protein-protein interactions. In this study, they address miRNA synergy in regulating expression of genes encoding uniformly expressed proteins (UEPs) and disorderly expressed proteins (DEPs) by employing experimentally validated and putative miRNA target sites, protein expression data in a panel of human organs and performing cell based assays with miRNA mimics. The research questions are well defined and relevant to the field. The authors use state of the art system biology tools, previously developed methods for calculating miRNA synergy scores and an appropriate choice of stringent statistical tests.
• While the classification of proteins into UEPs and DEPs and the skewness analysis follow good practice, in my opinion there are three major issues that should be addressed by the authors before the manuscript can be accepted:
1) The protein expression data allow the authors to classify proteins into UEPs and DEPs.
Then, the authors integrate protein-protein interaction data with miRNA target sites to assess miRNA synergy. The authors consider a number of conditions that are necessary for possible miRNA synergy to occur: genes bearing miRNA target sites need to be expressed in a particular organ and their protein products need to interact. My major concern with this approach is not taking miRNA expression profiles into consideration. As authors correctly note, there are miRNAs that are not widely expressed but their expression is restricted only to one or several organs. MiRNA synergy cannot physiologically occur if particular miRNAs are not expressed in the analyzed organs. The strategy adopted by the authors therefore leads to some computational predictions which most likely do not occur in the human body. For instance, miR-1 and miR-206, two miRNAs with high miRNA synergy potential predicted by authors in all 12 human organs in Figure 5, have been experimentally shown in many studies to have their expression restricted to the cardiovascular system and certain caner types (please see Mitchelson and Qin World J Biol Chem. 2015 for a review) and therefore cannot affect miRNA regulation network in the other organs. I believe that this study would benefit from including miRNA expression data and miRNA synergy studies should be restricted to miRNAs that are expressed in the analyzed organs. MiRNA expression data in human organs are publically available – please see Ludwig et al. Nucleic Acid Res 2016.
2) There is no information in the method section how the modelling of altered miRNA expression on miRNA regulation was performed (results depicted in Figure 3A-D). What is the modelled fold change of the miRNA expression? Is it within physiological range of observed miRNA expression changes? Why synergistic interactions are supposed for all possible miRNA pairs, while it is unlikely to be true in the biological setting - e.g. as seen in Figure 3E in this manuscript not all miRNA pairs have strong synergy (synergy score at least 1.0)?
3) The concept of comparing the effects of miRNAs targeting UEPs and DEPs on the whole cell state by miRNA mimic transfection followed by cell viability assay is correct in principle. However, it is not well executed. Firstly, the choice of miRNAs – miR-21 and miR-133a is questionable. MiR-21 is a well-known miRNA that targets tumor suppressor genes and it was shown experimentally in many studies to control apoptosis (please see Buscaglia and Li Chin J Cancer 2011 for a review). Therefore, miR-21 mimic cannot be used in an unbiased manner in the assay measuring cell viability as proposed by the authors. Secondly, the authors should state what the control group represents in the assay. Ideally, it should represent cells transfected with a miRNA mimic with no homology to any known human miRNA to account for the possible cell stress induced by the transfection reagent and possible off-target effects of used mimics.
• Lines 72-74: the authors should explain why they used different fraction thresholds for different human organs.
• Lines 106: the sequences (with description of RNA modifications, if relevant) of used miRNA mimics should be included.

Validity of the findings

• Most of the presented data have appropriately chosen controls. My major concern regarding controls is the choice of miRNA mimics in the cell viability assay, as described above.
• The presented data and conduced statistical tests are well chosen. Most of conclusions stem directly from the presented results and previously published work and are well stated. Some conclusions should be rephrased as they appear to be too far-reaching, e.g.: ‘miRNAs was sufficient to cause…’ (line 152).
• Lines 142-143: the authors conclude that ‘the preference of miRNAs for UEPs is also obvious to observe (Figure 2B)’, while they state in the figure legend that ‘no significance was found between UEPs and DEPs’. The main text needs to be corrected.

Comments for the author

No comment

---

## Round 0.2 · Minor Revisions

· Academic Editor

Minor Revisions

The manuscript has improved considerably and there are only minor comments left to solve as reported in the detailed reviewers' reports.

Reviewer 3 ·

Basic reporting

The Manuscript by Chen et al. has been largely improved.
Minor comment:
1. To be precise the title should be changed:
MicroRNAs tend to synergistically control EXPRESSION OF genes encoding extensively-expressed proteins in humans
2. Consistently, line 27: “Consistent with this, miRNAs that mainly target UEP-encoding genes (...)” should be changes for: “Consistent with this, miRNAs that mainly target UEP-encoding mRNAs (...)” or “Consistent with this, miRNAs that mainly target the expression of UEP-encoding genes (...)” – please include this precise description throughout the manuscript.

Experimental design

no comment

Validity of the findings

no comment

Reviewer 4 ·

Basic reporting

The English language has improved but the manuscript still required additional editorial correction, ideally by a native English speaker. There are still numerous mistakes that affect readability of the manuscript. For example, lines 11-13: this sentence makes no sense; line 25: “would like to”; line 32: should be “rely on” instead of “reply on”.

Experimental design

The experimental design of the cell viability assay is now much improved. Also more detailed description of used computational analyses was added.

Validity of the findings

I would strongly encourage the authors to consider rephrasing conclusions drawn from Figure 5 and Figure 6. Figure 5 identifies a number of miRNAs with high potential to regulate UEPs and therefore having greater chances to impact cell behavior. These miRNAs should be clearly stated as they were in the first version of the manuscript. Cell viability assay (Figure 6) is only one way of measuring effects of miRNAs on the cell state and it cannot be excluded that the other examined miRNAs with high potential to regulate UEPs (miR-1, miR-155) can affect the cell state in another way. Moreover, the authors should include the information of miR-21 being a well-known miRNA targeting tumor suppressor genes and therefore the results of the cell viability assay are in agreement with previous studies.

---

## Round 0.3 · accepted · Accept

· Academic Editor

Accept

The manuscript has been nicely improved and all the previous concerns have been addressed.